# Dendrite fragmentation induced by massive-like δ–γ transformation in Fe–C alloys

Hideyuki Yasuda [1], Kohei Morishita [1,5], Noriaki Nakatsuka [2,6], Tomohiro Nishimura[1,7], Masato Yoshiya [2], Akira Sugiyama [3], Kentaro Uesugi [4] & Akihisa Takeuchi [4]

Dendrite arm fragmentation is considered in solidification structure tailoring. Time-resolved and in situ imaging using synchrotron radiation X-rays allows the observation of dendrite arm fragmentation in Fe–C alloys. Here we report a dendrite arm fragmentation mechanism. A massive-like transformation from ferrite to austenite rather than the peritectic reaction occurs during or after ferrite solidification. The transformation produces refined austenite grains and ferrite–austenite boundaries in dendrite arms. The austenite grains are fragmented by the liquid phase that is produced at the grain boundary. In unidirectional solidification, a slight increase in temperature moves the ferrite–austenite interface backwards and promotes detachment of the primary and secondary arms at the δ–γ interface via a reverse peritectic reaction. The results show a massive-like transformation inducing the dendrite arm fragmentation has a role in formation of the solidification structure and the austenite grain structures in the Fe–C alloys.

[1] Department of Materials Science and Engineering, Kyoto University, Sakyo, Kyoto 606-8501, Japan. [2] Department of Adaptive Machine Systems, Osaka University, Suita, Osaka 565-0871, Japan. [3] Department of Mechanical Engineering for Transportation, Osaka Sangyo University, Daito, Osaka 574-8530, Japan. [4] Japan Synchrotron Radiation Research Institute (JASRI/SPring-8), Sayo-cho, Hyogo 679-5198, Japan. [5]Present address: Department of Materials Science and Engineering, Kyushu University, Nishi-ku, Fukuoka 819-0395, Japan. [6]Present address: Melting Section, Manufacturing Department, Moka Plant, Aluminum and Copper Business, Kobe Steel Ltd, 15 Kinugaoka, Moka, Tochigi 321-4367, Japan. [7]Present address: Kobe Corporate Research Laboratories, Kobe Steel Ltd., 1-5-5 Takatsukadai, Nishiku, Kobe, Hyogo 651-2271, Japan. Correspondence and requests for materials should be addressed to H.Y. (email: yasuda.hideyuki.6s@kyoto-u.ac.jp)

An understanding of the morphological transition from columnar to equiaxed structures during the solidification of metallic alloys is fundamental to control the micro-structures and grain structures of cast products. The detachment of fragmented dendrite arms is one mechanism that promotes the transition from a columnar to an equiaxed structure. Several factors can initiate dendrite arm fragmentation. One is remelting at the dendrite arm necks during coarsening between developing arms[1–6]. Another is remelting of dendrite arms after recalescence during solidification from an undercooled melt[7,8]. Both factors originate from shape instabilities that are associated with the dendritic geometries. Thus, dendrites have inherent shape instabilities that are driven by the solid–liquid interfacial energy; these are essentially the same as Plateau–Rayleigh instabilities. Another contributing mechanism is local solute enrichment, which is caused by melt flow in the mushy region[1,9–14]. Depending on the configuration of the dendrite arms and the temperature distribution, melt flow in the mushy region can cause local solute enrichment around the necks of dendrite arms and promotes fragmentation. Radiography of the solidification in metallic alloys has shown that fragmentation that is caused by local solute enrichment can occur spontaneously in the mushy region[9–14]. All known fragmentation mechanisms are based on shape instability that is caused by the curvature effect (the capillary effect). Pinch-off of dendrite arms as a result of this effect has therefore been synonymous with fragmentation for the last few decades.

Solid grain isolation by liquid film occurs at temperatures below the liquidus temperature but above the solidus temperature, if the grain boundary energy is more than double the solid–liquid interfacial energy. This condition is satisfied in various alloys[15–18]. In addition, pre-melting at the grain boundary below the melting temperature[19–24] has also been reported. According to the interfacial energies in the literatures[25–29] and the observed microstructure of semisolids[30–32], Fe–C alloys should satisfy the condition. However, melting at the grain boundary does not occur in normal dendritic solidification, because no grain boundary exists in growing dendrite arms. Thus, the melting at the grain boundary has not been included into the dendrite arm fragmentation mechanism.

In Fe–C alloys with carbon contents < 0.5 mass%, the γ phase (austenite, face-centered cubic) is considered to be produced through a peritectic reaction between the δ phase (ferrite, body-centered cubic) and the liquid phase. Casting defects, such as cracking during the continuous casting of carbon steel have been discussed with the peritectic reaction[33–36]. Volume changes from the peritectic reaction are considered as a source of strain in the solidifying shell[33,34,37–39]. Thus, an understanding of the transformation from the δ phase to the γ phase is interesting from a scientific and industrial perspective. Radiography has enabled the observation of solidification phenomena (dendritic growth, δ–γ transformations, and semisolid deformations), even in Fe-based alloys[30,40–43]. Observations of X-ray imaging[42–44] and laser-scanning confocal microscopy[45] have proven that a massive-like transformation, in which the δ phase is undercooled into the δ + γ region or the single γ region in the equilibrium phase diagram and transformed to the γ phase, is selected during and after solidification in the Fe–C alloys.

In this work, we demonstrate dendrite arm fragmentation that is induced by a massive-like transformation from the δ phase to the γ phase. Time-resolved and in situ observations using synchrotron radiation X-rays prove that dendrite arm fragmentation occurs at the γ grain boundary and the δ–γ interface after a massive-like transformation. A massive-like transformation at 20 K below the peritectic temperature produces fine γ grains in dendrite arms in an Fe–0.45 mass% C–0.6 mass% Mn–0.3 mass% Si alloy (0.45 C steel). The liquid film at the γ grain boundary is developed when temperature is kept constant and consequently the dendrite arms are fragmented. In the unidirectional solidification in Fe–0.58 mass% C–0.6 mass% Mn–0.3 mass% Si (0.58 C steel), backward motion of the δ–γ interface from the reverse peritectic reaction induces multiple dendrite arm fragmentations. A massive-like transformation rather than a peritectic reaction is involved in formation of the solidification structure and austenite grain structures in the Fe–C alloys. Therefore, this study provides a fundamental understanding of the solidification microstructure and associated casting defect formation in the peritectic Fe–C alloys.

## Results

**Fragmentation induced by melting of γ grain boundary.** Figure 1 shows transmission images during solidification of 0.45 C steel. The observations, in which the pixel size was 1 μm × 1 μm,

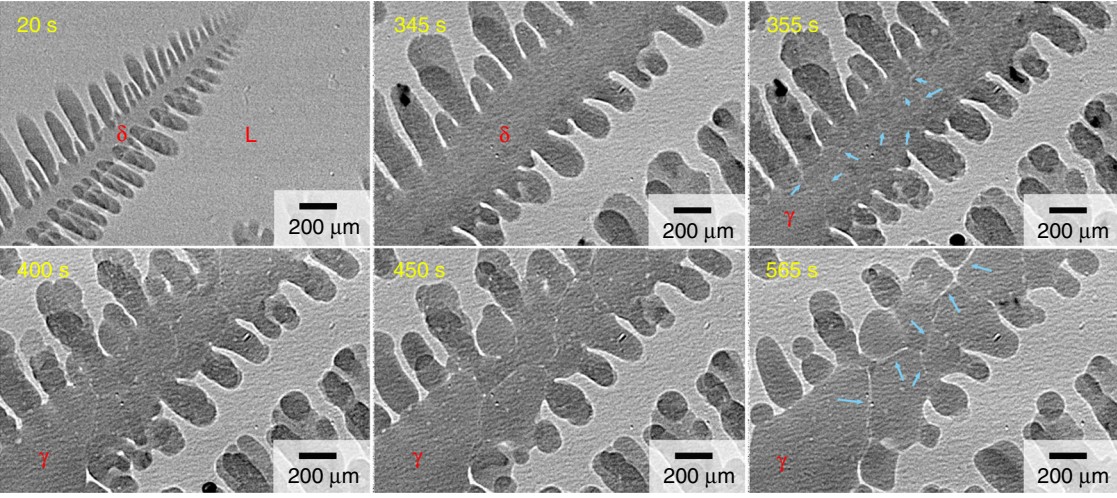

**Fig. 1** Fragmentation of γ grains induced by a massive-like δ–γ transformation in 0.45 C steel. The melted specimen was initially cooled at 0.17 K s$^{-1}$ and held at 20 K below the peritectic temperature after 152 s. A δ dendrite arm grew from the bottom-left corner to the top. The massive-like δ–γ transformation occurred at 346 s. Blue arrows indicate the liquid film that was produced at the γ grain boundary after the massive-like transformation. Liquid films at the γ grain boundaries were thickened after the massive-like transformation. The X-ray energy and exposure time were 21 keV and 50 ms, respectively

were performed at the BL20XU at SPring-8 (Hyogo, Japan) to investigate melting of the γ grain boundaries after a massive-like transformation. The specimen was cooled initially at $0.17\,K\,s^{-1}$ under a temperature gradient of $2 \times 10^3\,K\,m^{-1}$ in the vertical direction. Dendrites of the δ phase (δ dendrite) grew from the left–bottom to the right–top. Nucleation of the γ phase (γ nucleation) did not occur at or below the peritectic temperature, and hence, δ dendrites continued to grow below the peritectic temperature. The temperature was held at ~ 20 K below the peritectic temperature (after 152 s) and the temperature gradient was maintained. At this stage, the width of the primary arms was as large as 400 μm and the secondary arms continued to coarsen during the temperature holding.

The massive-like δ–γ transformation occurred at 346 s. The liquid phase in equilibrium with the δ phase at temperatures below the peritectic temperature was undercooled below the liquidus temperature of the γ phase, and thus, the γ phase with a perturbed liquid (L)–γ interface grew rapidly, as shown in the image at 355 s. Dark regions, where the Bragg condition was satisfied, were observed in the γ phase. The probability that a grain satisfies the Bragg condition is low because of the low divergence of $> 10^{-4}$ rad and the X-ray energy resolution ($\Delta E/E$) of $10^{-4}$ at the beamline, and hence, the presence of many dark regions indicates that fine γ grains with grain boundaries between them were formed during the massive-like transformation. In general, the formation of dendrite arms that consist of multiple grains is rare in solidification processes.

A film-like liquid phase (liquid film), which is indicated by blue arrows, was partially formed at γ grain boundaries. The liquid film was not clearly visible immediately after the massive-like transformation, as shown in the image at 355 s. The liquid film actively migrated because of the coarsening and thickening of γ grains, as shown in the transmission images after 400 s. The supplemental movie shows that the migration velocities of the liquid film at the γ gain boundaries were as fast as $1\,\mu m\,s^{-1}$ (see motion of blue circle in the movie). As a result of this coarsening, the γ grains in the dendrite arms were isolated by the liquid phase, as shown in the image at 565 s. However, detachment of fragmented γ grains was not observed. The sample confinement in the specimen holder with a thickness of 100 μm and/or a relatively large viscous drag force owing to the thin liquid film at the γ grain boundaries can inhibit detachment. Here, we comment on the limitation of this observation (X ray transmission imaging). Significant technological improvements in time-resolved tomography, which are applicable for the three-dimensional observation of steel dendrites with high spatial and temporal resolutions, are required to obtain further evidence of whether the fragmented arms are detached spontaneously or not within this time frame.

The fragmentation was not observed after a massive-like transformation when the specimen was cooled continuously[42–44]. It is worth noting that fragmentation occurs everywhere in the dendrite arms only when the temperature was kept below the peritectic temperature after a massive-like transformation, as will be discussed later.

**Multiple fragmentation induced by a reverse peritectic reaction.** Multiple fragmentation at the δ–γ interface, which was induced by the massive-like transformation, was also observed during columnar dendritic growth at a higher carbon content, i.e., 0.58 C steel. Although the γ phase is the primary phase and no transformation from δ to γ phases is expected at equilibrium, preferential nucleation and solidification of the metastable δ phase was often observed in 0.58 C steel. Observations were performed across a relatively wide area at the BL20B2. Figure 2

shows transmission (left) and differential (right) images during solidification and the massive-like transformation in 0.58 C at a cooling rate of $0.17\,K\,s^{-1}$. A differential image at $t = t_0$ was obtained by subtracting the transmission image at $t = t_0 - 10$ s from the transmission image at $t = t_0$.

Columnar δ dendrites, which were tilted away slightly from the direction of the temperature gradient (vertical direction), grew from the bottom to the top. The front of the massive-like δ–γ transformation was observed behind the dendrite tips, as shown in the image at 0 s. The red lines indicate the growth front of the massive-like transformation. The distance between the tips and the growth front was ~ 2 mm at 0 s. In the differential image at 0 s, the dark regions indicated by blue arrows correspond to the growth lengths of the δ and γ phases during the last 10 s. The growth velocity of the γ phase was slightly higher than that of the δ dendrites, and thus γ phase growth was likely to catch up with δ dendritic growth. Dendrite arm fragmentation was never observed during nearly steady growth of columnar dendrites, which supports the case for continuous cooling.

Cooling was suspended at 14 s and the temperature was maintained. Because of the delayed response of the temperature control, the temperature continued to decrease slightly and then increased towards the controlled temperature. Almost no contrast was observed in the differential image at 20 s, which shows that δ and γ phase growth stopped temporarily in the last 10 s. Then, the δ–γ interface moved backwards and multiple fragmentation occurred spontaneously and simultaneously. A coupling of the light (left) and dark (right) contrasts in the differential image at 39 s indicates that the primary dendrite arms, which were fragmented at the δ–γ interface, leaned to the right because of the gravitational force. The backward motion of the δ–γ interface triggered detachment of fragmented arms. Formation of a branched columnar grain structure, which is typically formed in the intermediate zone from columnar to equiaxed grain zones in solidification structures in carbon steel[46], is explained by considering the tilted dendrite arms caused by the multiple fragmentation.

Figure 3 shows close-up views of dendrite arm fragmentation in 0.58 C steel. As shown in Fig. 3a, fragmentation at the root of a secondary arm, which is indicated by red arrows between two green lines, occurred at 39 s. Most of the primary dendrite arms in the observation area also fragmented at the δ–γ interface, as shown in Fig. 3b. The solid–liquid interface was traced to identify the shape of fragmented interface. The blue and red lines indicate the solid–liquid interface at 33 s (before fragmentation) and 60 s (after fragmentation), respectively. A previous study showed that in fragmentation that is driven by the solid–liquid interfacial energy[3], pinching resulted in fragmentation; the typical shape of the fragmented arms is shown in Fig. 3d. However, in this study, no pinching was observed in the fragmented arms, as shown in Fig. 3b. The primary arm was fragmented at the δ–γ interface, as if it had been cut by a blade, as shown in Fig. 3c. Fragmentation without pinching at the δ–γ interface was observed in Figs. 2 and 3, and is consistent with melting at the γ grain boundary in 0.45 C as shown in Fig. 1.

**Melting of γ grain boundary.** Melting of the γ grain boundary occurs to reduce the total interfacial energy. If the relationship $\sigma_{\gamma/\gamma} \geq 2\sigma_{\gamma/L}$, where $\sigma_{\gamma/\gamma}$ and $\sigma_{\gamma/L}$ are the γ boundary energy and γ–liquid interfacial energy, respectively, is satisfied, a liquid film will form at the γ grain boundaries, even though the γ–liquid interfacial area is twice that of the γ–γ interfacial area. Values for the γ boundary energy in the alloy at high temperatures have not been reported in the literature. The γ grain boundary energy for γ-Fe that is saturated with Cu and S was estimated to be $0.85\,J\,m^{-2}$ [25]. The average grain boundary energies at $T/T_m = 0.7$, 0.8, and 0.9 were

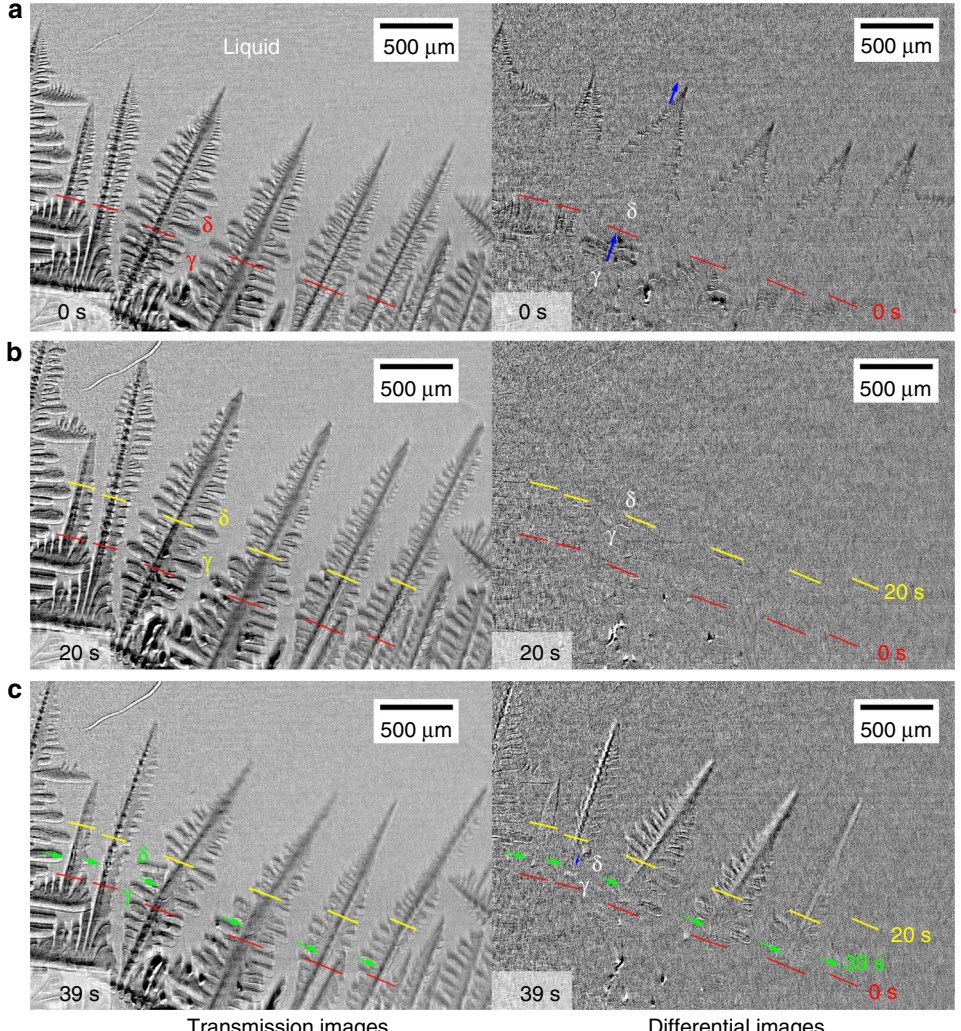

**Fig. 2** Multiple fragmentation in the unidirectional solidification of 0.58 C steel. Left and right images are transmission and differential images, respectively. Differential images were obtained by subtracting a transmission image obtained 10 s earlier from a transmission image. **a**, **b** δ dendrite arms grew from the bottom to the top. Red and yellow lines indicate the δ–γ interface position at 0 s **a** and 20 s **b**, respectively. **c** Green arrows indicate the δ–γ interface when multiple fragmentation occurred and leaned to the right at 39 s **c**. Blue arrows approximately indicate growth/melting length in the last 10 s. The X-ray energy and exposure time were 20 keV and 1 s, respectively

estimated to be 0.6, 0.8, and 1.0 J m$^{-2}$, respectively[26]. Here, $T_m$ is melting temperature. Atomistic simulations that were performed by the embedded atom method gave γ grain boundary energy ranges from 0 to 0.4 J m$^{-2}$, depending on the degree of misorientation[27]. The phase field model gave an average γ grain boundary energy of 0.37 J m$^{-2}$ [28]. The solid–liquid interfacial energy for iron has been reported to be 0.2 J m$^{-2}$ [29]. Therefore, it appears reasonable that the relationship $\sigma_{\gamma/\gamma} \geq 2\sigma_{\gamma/L}$ can be satisfied for Fe–C alloys, although it is still difficult to compare the interfacial energies quantitatively.

In addition to the estimated interfacial energies, time-resolved and in situ observations of semisolid deformation in Fe–C alloys[30–32] proved qualitatively that the relationship $\sigma_{\gamma/\gamma} \geq 2\sigma_{\gamma/L}$ is satisfied in the Fe–C alloy system. The observations show that solid grains were always isolated by a liquid phase, even at a solid fraction of 0.9, and the rearrangement of isolated grains played a dominant role in controlling semisolid deformation. The isolation of grains should not be observed unless the relationship $\sigma_{\gamma/\gamma} \geq 2\sigma_{\gamma/L}$ is satisfied.

The time-resolved and in situ observations showed that solid grains in Al–Cu alloys were also isolated by a liquid film at high solid fractions[47,15]. The solid–liquid interfacial energies and high-

angle grain boundary energies for Al–Cu, Al–Si, and Pb–Sn alloys were determined precisely from the shapes of their grain boundary cusps[16]. The measured values of $2\sigma_{S/L}/\sigma_{S/S}$, where $\sigma_{S/L}$ and $\sigma_{S/S}$ are the solid–liquid interfacial energies and the grain boundary energy, respectively, are approximately equal to one for these alloys. Recently, time-resolved tomography showed that transgranular fractures could occur in the semisolid state of Al–Cu alloys and result in liquation cracking[17,18]. Therefore, the results support melting at grain boundary, not only in Fe–C alloys but also in other alloy systems.

It is of interest to consider pre-melting at the grain boundary below the melting temperature[19–22]. From a thermodynamics perspective, the driving force of pre-melting at the γ grain boundary is essentially given by the relationship, $2\sigma_{\gamma/L}-\sigma_{\gamma/\gamma}$[19]. Tensile tests showed that pre-melting at the grain boundary occurs at several kelvins below the melting temperature in high-purity aluminum[20]. However, no experimental evidence of pre-melting at the grain boundary for pure aluminum was observed by in situ TEM observations[21]. As mentioned in previous work[21], pre-melting can be regarded as atomistic disordering process at the grain boundary at lower temperatures, and melting at

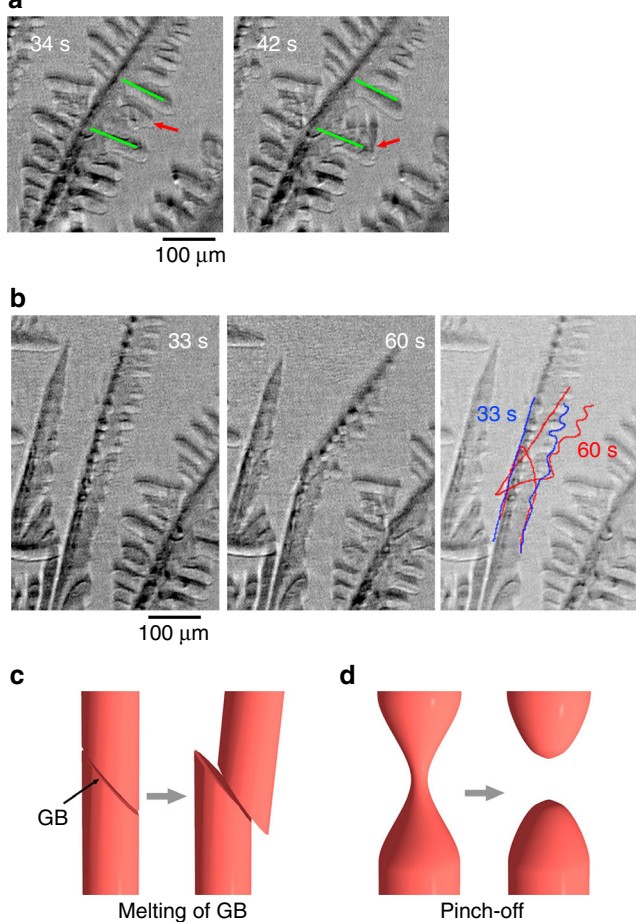

**Fig. 3** Close-up views of fragmentated arms. **a** Fragmentation of the secondary dendrite arm in 0.58 C steel. The secondary arm that is indicated by a red arrow between the secondary arms that are indicated by green lines was fragmented between 34 s and 42 s and settled down because of the gravitational force. **b** Fragmentation of the primary dendrite arms in 0.58 C steel. The fragmented arm was tilted because of the gravitational force. Blue and red lines indicate the solid–liquid interface at 33 s and 60 s, respectively. **c** Schematic diagram of dendrite arm fragmentation at the γ grain boundary and the δ–γ interface in Fe–C alloys. **d** Schematic diagram of fragmentation caused by the pinch-off

the grain boundary occurs abruptly close to the melting temperature. Recently, atomistic simulations showed that a grain boundary becomes disordered atomically and is even transformed into a thin liquid film at temperatures near the melting temperature[23,24]. Thus, the liquid film thickness at the grain boundary should be small compared with the spatial resolution of the X-ray imaging. The arguments indicate that a massive-like transformation did not induce the thick liquid film at the γ gain boundary if the transformation occurred in the δ phase after solidification.

**Development of liquid film at γ grain boundary.** Fragmentation of dendrite arms was observed only when the temperature was kept constant or increased slightly, as mentioned above. In addition, the liquid films at the γ grain boundaries were thickened during coarsening, as shown in Fig. 1. The development of a liquid film at the γ grain boundaries is explained by considering carbon diffusion after a massive-like transformation in the Fe–C binary system. Figure 4 shows the carbon concentration profiles before and after the massive-like transformation with respect to

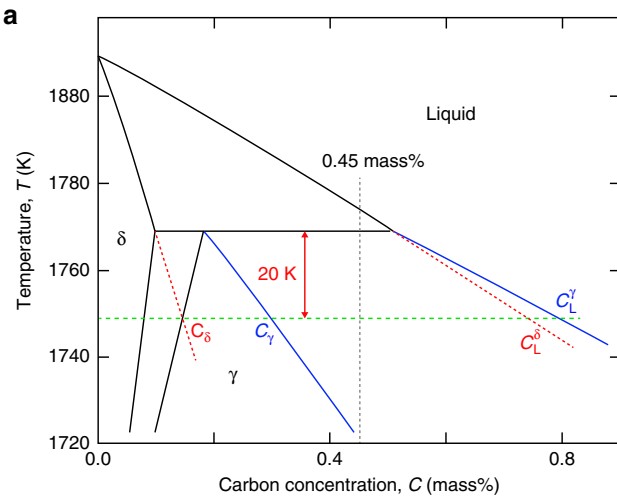

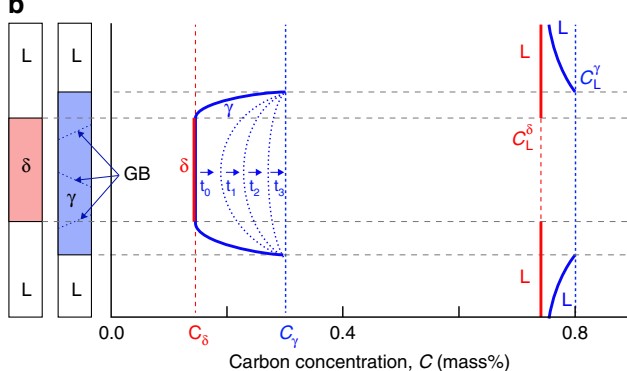

**Fig. 4** Diffusion-controlled melting at γ grain boundary. **a** Phase diagram of Fe–C alloy system. A massive-like transformation occurred at 20 K below the peritectic temperature (dashed red line). **b** Carbon concentration profiles before (red line) and after (blue lines) massive-like transformation. Carbon atoms diffuses from the liquid–γ interface to the core of the dendrite arm. Limited liquid phase formed until the carbon concentration in the γ phase became $C_\gamma$, in equilibrium with the liquid phase

the Fe–C phase diagram, which allows us to consider the behavior from a thermodynamic perspective. The massive-like transformation in Fig. 1 occurred at ~20 K below the peritectic temperature [red dashed line in Fig. 4a]. The left part of Fig. 4b depicts the one-dimensional configuration of the liquid and δ phases at 20 K below the peritectic temperature, and the carbon concentration profile at equilibrium, $C_\delta$, is shown by red solid lines on the right of Fig. 4b. The carbon concentration in the δ phase almost falls into the single γ phase region in the equilibrium phase diagram, and hence transformation from δ to γ phases can occur thermodynamically without carbon partitioning at the δ–γ growth front. If a γ phase is formed without carbon partitioning, the carbon concentration in the γ phase should be equal to $C_\delta$ immediately after the massive-like transformation. Simultaneously, the γ phase will grow rapidly from the L–γ interface that is produced by the massive-like transformation because the liquid phase is undercooled below the liquidus temperature of the γ phase and now is in intimate contact with the γ phase. As a result of the massive transformation (without carbon partitioning) and rapid growth, the carbon concentration immediately after the massive-like transformation will be as per the profile that is indicated by the blue solid lines in the schematic diagram in Fig. 4b. After the massive-like transformation, the carbon concentration at the core of the primary dendrite arms (γ phase) increases toward the carbon concentration in

equilibrium with the liquid phase, $C_\gamma$, through carbon diffusion from the L–γ interface to the core in the γ phase.

Liquid films do not form at the grain boundary of a γ phase with a carbon content below $C_\gamma$ because equilibrium between the γ and liquid phases cannot be achieved. Therefore, an increase in the carbon content by further diffusion from the liquid phase to the dendrite core is needed for melting at grain boundaries. Carbon diffusivity in the γ phase was estimated to be of the order of $10^{-10}$ or $10^{-9}$ m$^2$ s$^{-1}$ [48,49]. The diffusion length, which is defined as the square root of $2Dt$ ($D$ is the diffusivity and $t$ is the duration), was of the order of 100 μm for $t = 100$ s. The observed width of the dendrite arms ranged from 50 to 400 μm, and thus it probably takes 10–100 s to achieve a nearly uniform carbon concentration in the γ phase. The development of a liquid film after the massive-like transformation, as observed in Fig. 1, is consistent with this consideration.

**Detachment of fragmented dendrite arms**. The maximum thickness of the liquid film was 10 μm at 565 s and no detachment was observed (Fig. 1). Detachment of fragmented arms needs melt flow into the gap between γ grains, and the pressure drop at the gap causes attractive forces between the fragmented arms. In addition, sample confinement in the specimen holder could inhibit detachment. The experimental results showed that the gravitational and buoyancy forces were too small to detach the fragmented arms if the width was 10 μm. Detachment was observed when the δ–γ interface moved backwards, as shown in Fig. 2. In the backward motion, a reverse peritectic reaction, γ→δ+L, produced the liquid phase and consequently, thickened the liquid film at the δ–γ interface. The increases in thickness of the liquid film significantly decrease the attractive force between the fragmented arms, and therefore detachment can occur easily. The detachment that is associated with the backward motion suggests that a slight increase in temperature during solidification promotes detachment of fragmented dendrite arms at the δ–γ interface, which leads to branched columnar[46] and equiaxed grains that are observed in solidification structures of carbon steels. The detachment is expected in hyperperitectic compositions rather than in hypoperitectic compositions because the solid fraction at the δ–γ interface decreases with an increase in carbon content.

**Dendrite arm fragmentation mechanism**. Here, the fragmentation mechanism that was observed in this study is compared with several other mechanisms that are already known. One mechanism is the temperature gradient zone melting (TGZM)[4], which was confirmed by radiography[5,6] Li et al.[5] reported that solute transfer in the mushy region at a temperature gradient from 0.7 to 2.4 K mm$^{-1}$ promoted dendrite fragmentation in Sn–13 mass% Bi alloys. Thi et al.[6] also proved that fragmentation due to TGZM occurred in Al–3.5 mass% Ni and Al–7.0 mass% Si and the migration velocity (0.25–0.31 μm s$^{-1}$) that was measured by the observation well agreed with the TGZM model. Solute transport owing to melt flow in the mush was confirmed as another mechanism of dendrite fragmentation[11,14] In both mechanisms, solute transport from diffusion in the liquid phase and/or melt flow controls melting at the dendrite arm necks. During melting at the neck owing to local solute enrichment, some dendrites are fragmented, depending on the configuration of the dendrite arms and cooling conditions, which means that fragmentation events can occur accidentally in specific dendrite arms. The other mechanism is the shape instability mechanism, which requires a certain degree of undercooling before solidification for fragmentation during the temperature plateau after recalescence[7]. In contrast, in the mechanism of melting at the δ–γ

interface, the interfaces in all dendrite arms move backwards in the same way, which leads to multiple fragmentation. The backward motion follows the temperature change in every dendrite arm, and thus multiple fragmentation events occur in the dendrite arms at the same isothermal plane and can be controlled by precise cooling and heating in accordance with the phase equilibrium. Although transformation from δ to γ phases could contribute to grain refinement in the Fe–C alloys, a scientific understanding for the refinement has remained unclear[50]. This study demonstrates how fragmentation occurs during solidification of Fe–C alloys. Multiple fragmentation control by cooling and heating is an attractive method for achieving grain refinement.

**Massive-like transformation from current and future perspectives**. It is critical from scientific and industrial perspectives to study a massive-like transformation as opposed to the peritectic reaction. Cracking in hypoperitectic Fe–C alloys during continuous casting has been discussed with the peritectic reaction[33,34]. A critical strain has been proposed to predict the formation of internal cracking in continuous casting[35,36]. According to the studies[35,36] the critical strain ranged from 1 to 4%, depending on strain rate and carbon content. Recently, volume shrinkage in the transformation from δ to γ phase was measured by time-resolved tomography[51]. Because the measured volume shrinkage was ~ − 0.5%[51], the strain that was induced by isotropic shrinkage causes a strain of only − 0.17%. Thus, the strain that was locally induced by a massive-like transformation is not linked directly to crack formation at the liquid film between the γ grains. Non-uniform strains that are induced by various origins in real casting processes must be considered (i.e., massive-like transformation, friction between the solidifying shell and mold, unevenness of the solidifying shell, and static pressure)[33,34,37–39]. Future challenges include a massive-like transformation as one of the strain sources and an understanding of the influence on the non-uniform strain in the solidifying shell. In addition, the response of γ grains that were separated by the liquid phase to the strain and stress must be examined.

In peritectic systems, a high-temperature phase will be undercooled easily into the co-existence region of the high- and low-temperature phases if the interface between the high-temperature and liquid phases is not a preferred nucleation site for the low-temperature phase. As a result, a solid transformation from the high- to low-temperature phases can be selected by a mechanism similar to that associated with dendrite arm fragmentation in Fe–C alloys. Recently, a massive-like transformation was also observed in Fe–Cr–Ni alloys[52]. Therefore, it is of interest to examine the solid transformations during solidification in Ti–Al, Cu–Fe, and other alloys with peritectic reactions.

Although the grain boundaries are formed by a massive-like transformation in this work, the presence of grain boundaries in a single dendrite arm rather than a massive-like transformation is a necessary condition for multiple fragmentation. Other possibilities exist for forming grain boundaries in a single dendrite arm. For example, liquation cracking[17,18] indicated distinctly that recrystallization, which was caused by deformation of solid grains and subsequent heating to the semisolid region can also achieve refined grain formation in a single dendrite arm. Therefore, multiple fragmentation, which is found in Fe–C alloys, could be used to improve grain structures in other metallic alloys.

## Methods
**Specimens**. The compositions of carbon steels with carbon contents of 0.45 mass% and 0.58 mass%, which were produced by a conventional steel-making method, were determined by chemical analysis. The compositions of the steels used for observations were Fe–0.45 mass% C–0.73 mass% Mn–0.2 mass% Si, and Fe–0.58

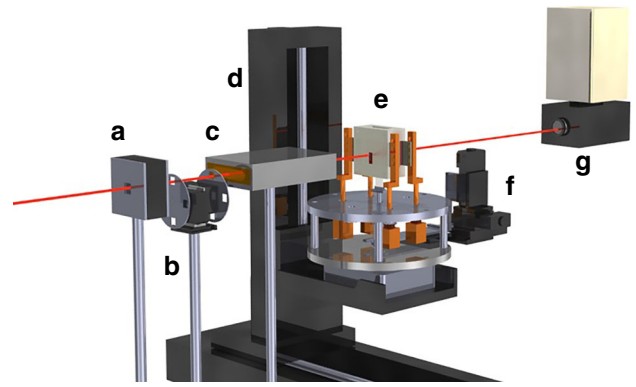

**Fig. 5** Setup for time-resolved and in situ observations. **a** Four-jaw slit to define the incident beam size. **b** Absorber (a $SiO_2$ glass plate) to adjust the intensity of the X-ray beam. **c** Ion chamber to measure the intensity of the X-ray beam that irradiates a specimen. **d** Stages to control furnace position. **e** Specimen and furnace that consist of a graphite heater and BN frames. The specimen is heated under a vacuum chamber. **f** Stages to control specimen position. **g** X-ray beam monitor to observe X-ray transmission images. Red line indicates an X-ray beam (20 or 21 keV)

mass% C–0.6 mass% Mn–0.3 mass% Si. Solidification of the δ phase was expected to be followed by peritectic solidification at equilibrium in 0.45 C. Solidification of the γ phase was expected to be completed without a δ phase in 0.58 C.

A specimen of 5 mm width, 10 mm height, and 0.1 mm thickness was inserted between two sapphire plates of 150 μm thickness and solidified in a specimen holder at a cooling rate from 0.17 to 0.5 K s$^{-1}$ under a vacuum of less than 10 Pa. Sample confinement in the specimen holder could influence the microstructure evolution because of a limited diffusion field. Solid grain motion was restricted, and the solid grains could be piled up in the thin space. Thus, it is plausible to assume that the confinement could hinder detachment of fragmented arm. However, the selection of a massive-like transformation and the induced fragmentation were not suppressed by confinement under the geometric conditions in this study.

The interfacial energy of the solid–liquid phases for pure iron, $\sigma_{S/L}$, was reported to be 0.2 J m$^{-2}$ [53]. The value of $\sigma_{Sapphire/S}$–$(\sigma_{S/L}+\sigma_{Sapphire/L})$ was calculated to be + 0.2 J m$^{-2}$ [54]. Here, $\sigma_{Sapphire/S}$ and $\sigma_{Sapphire/L}$ are the sapphire-solid and the sapphire-liquid interfacial energies, respectively. The positive value indicates that the solid phase does not stick to the sapphire plates, and the liquid phase tends to exist between the solid phase and the sapphire plate during solidification. Therefore, the friction between the solid grain and the sapphire plate is expected to be relatively small. As shown in Fig. 3, the detachment of dendrite arms occurred when the solid grains were sufficiently isolated from each other by the liquid phase and the solid fraction was sufficiently low.

**X-ray imaging**. X-ray transmission imaging was performed at beamlines BL20B2 and BL20XU at SPring-8 (a synchrotron radiation facility in Hyogo, Japan). The time-resolved and in situ observation techniques were essentially the same as those used in previous studies[40,41,30,42–44] Figure 5 shows the experimental setup used in this study[41]. A monochromatized X-ray beam of energy 20 or 21 keV was used for transmission imaging. A four-jaw slit for defining the incident beam size, an absorber (a $SiO_2$ glass plate) to adjust the X-ray beam intensity, an ion chamber to measure the intensity of the X-ray beam that irradiates a specimen (if needed), a furnace that consists of a graphite heater and BN frames, and an X-ray beam monitor to observe X-ray transmission images were placed along the X-ray beam. Beam monitors with pixel sizes of 1 μm × 1 μm at the BL20XU and 4 μm × 4 μm at the BL20B2 were used and the typical frame rate for the transmission images was 1 fps at both beamlines. The image signals were converted to a digital format with 12- or 13-bit resolution.

**Image processing**. We used conventional image processing to improve the image quality. The intensity of transmission X-ray, $I$, is a function of position, $(x, y)$, and time, $t$. The intensity of the X-rays that were transmitted though the specimen and specimen holder, $I_{S+C}$, is expressed as:

$$I_{S+C}(x, y, t) = I_0(x, y) \exp[-\mu_C d_C] \exp[-\mu_S(x, y, t) d_S(x, y, t)]. \quad (1)$$

where $\mu$ and $d$ are the linear absorption coefficient and thickness, respectively. The subscripts S and C indicate specimen and specimen holder, respectively. $I_0$ is the intensity of incident beam. The intensity of X-rays that were transmitted through the uniform liquid phase before cooling, $I_{L+C}$, is given by:

$$I_{L+C}(x, y, t) = I_0(x, y) \exp[-\mu_C d_C] \exp[-\mu_L d_L(x, y)], \quad (2)$$

Where the subscript L indicates the liquid phase. The transmitted images ($I_{S+C}$)

were normalized in terms of the image that was transmitted though the liquid phase ($I_{L+C}$):

$$I_N(x, y, t) = I_{S+C}/I_{L+C} = \exp\{-[\mu_S(x, y, t)d_S(x, y, t) - \mu_L d_L(x, y)]\} \quad (3)$$

Normalization removes the effects of a non-uniform distribution of the incident X-ray beam and the effects of the specimen holder. The normalized images were of sufficient quality to enable detection of the solid–liquid interface.

The difference between the densities of the δ and γ phases was less than 1%, and thus it was difficult to detect the front of the δ–γ interface. To enhance the difference between the frames, differential images, $I_D(x,y,t)$, as described by Equation 4, were also obtained.

$$I_D(x, y, t) = I_{S+C}(x, y, t)/I_{S+C}(x, y, t - \Delta t) \quad (4)$$

The differential images enabled detection of the growing front of the δ–γ transformation although the density difference was < 1%.

**Data availability**
All data that support the findings of this study are available in this paper and the supplementary movies, or they are available from the corresponding author on reasonable request.

**Code availability**
Computer code used in the image processing is nothing more than Equations (1–4), which can be done by a calculator in principle.

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

## Acknowledgements

Observations at the synchrotron radiation facility SPring-8 were performed as general projects (2011B1247, 2012A1269, 2016A1454, 2017A1336, and 2018B1521) at BL20B2 and BL20XU of SPring-8 (JASRI), Japan. H. Yasuda, K. Morishita, and M. Yoshiya acknowledge financial support from the research group Visualization of Solidification of the 19th Committee on Steelmaking of the Japan Society for the Promotion of Science, and from Heterogeneous Structure Control: Towards Innovative Development of Metallic Structural Materials of the Industry–Academia Collaborative R & D Program (JST). The X ray transmission imaging that was developed with funding from a Grant-in-Aid for Scientific Research (S) (No. 17H06155) allowed the authors to observe solidification phenomena in Fe-based alloys. We thank Dr. T. Nagira, from Joining and Welding Research Institute, Osaka University, for his support during the initial stages of this study. We also thank Helen McPherson, PhD, and Laura Kuhar, PhD, from Edanz Group (www.edanzediting.com/ac) for editing a draft of this manuscript.

## Author contributions

H.Y. conceived the work and designed the high-temperature X-ray observation apparatus. K.M., N.N., T.N, A.S. and H.Y. developed the apparatus for the high-temperature observations and performed X-ray imaging experiments at SPring-8. H.Y. processed the X-ray transmission images. M.Y. and H.Y. analyzed the fragmentation and determined the fragmentation mechanism. K.U. and A.T. (beamline scientists) contributed to the development of X-ray imaging and an optimization of optics to observe steel solidification at high temperature. H.Y. wrote the paper, with contributions from all authors.

## Additional information

**Competing interests:** The authors declare no competing interests.

**Peer Review Information:** *Nature Communications* thanks Charles-André Gandin and other anonymous reviewer(s) for their contribution to the peer review of this work. Peer reviewer reports are available.

