## [Transparent Peer Review File · Nature Communications]

Reviewers' comments:

Reviewer #1 (Remarks to the Author):

The submitted manuscript reports on new dendrite fragmentation mechanisms observed under massive like peritectic transformation and in a reversed peritectic reaction during solidification of Fe-C steels. Such mechanisms would be of major relevance for industries if they could be adapted in to processes promoting grain refinement and improved microstructures in the casting of steels. Although the governing mechanisms may not have been regarded previously as potential candidates to initiate dendrite fragmentation, they are not necessarily novel, and I would like to see the present study better linked with previous works. I also have some concerns with their potential for industrial applications, as both seem to require some form of interrupted cooling process and holding times, which could turn out very difficult to realize in practice.

As for mechanism no. 1, the massive-like peritectic transformation, I have some concerns with the impact of the sample confinement, and would like to see this issue further elaborated in order to judge whether this mechanism truly have detachment potential. For the second mechanism, the reversed peritectic reaction, the case is more evident. Since the two fragmentation mechanisms constitute the core of the submission, I have raised my main criticism in greater detail in the two in bullets 1) and 2) below. Some minor (technical) issues are raised in subsequent bullet points.

1) Concerning the first case, illustrated in Fig.1 and in the supplementary video sequence "melting_GB.mp4": The fragmentation, or crystal fracturing, is apparently initiated by stresses accompanying the massive-like delta-gamma transformation. This relates to existing models for high-temperature solidification cracking in Fe-C based alloys, where even non-massive diffusion controlled peritectic transformations have been suggested as a source for internal crack formation, dating back at least 3-4 decades (Brimacombe and Sorimachi, Met. Trans 8B, 1977, 489-505; Clyne, Wolf and Kurz, Met Trans 13B, 1982, 259-266). The authors for the current submission have published results previously, combining in situ radiography and XRD, demonstrating very prominent strain effects associated with massive like transformations (ref. 14 in the manuscript), and I am not convinced that the present case study no. 1 represents something distinctively new. As far as I can tell, a difference in the current situation, compared to the one covered in ref. 14 is the present interruption of the sample cooling, which allows the cracked gamma crystals to be better exposed to liquid. The holding time also allows the system to approach equilibrium and gives time for the carbon transport required to initiate preferential infiltration of liquid into cracks associated with higher grain-boundary misorientations, as clarified through the opening paragraphs of the discussion. The latter bear clear similarities to well-established hot tearing models. In sum, neither the crystal cracking nor the liquid infiltration mechanisms are genuinely new.

The pending issue is therefore if the cracking caused by massive-like delta-gamma transformation can lead to crystal fragmentation, and to me this remains unclear. It is evident both from images and supplementary material that the crystals break up into sub-regions with internal misorientation, evidenced by the extinction effects observed in radiography when separate sub-volumes satisfy the Bragg-condition. Secondly, the rather long incubation time required for liquid infiltration, on a minute scale after the delta-gamma transition, is well explained by the C-diffusion characteristics outlined in the discussion (bottom page 10 and top page 11). Yet, I would expect, if this model was correct, that liquid infiltration should occur first at cracks in secondary arms, where gamma C diffusion completes faster. I do not see any evidence in the video that clearly resolvable liquid films are formed in secondary branches, but rather that continuous thicker films are formed more or less exclusively in the primary trunks. At the same time, it is evident that fracturing has happened also inside secondary arms, both from visual extinction, as well as Ostwald ripening events occurring internally inside regions that originated as a single secondary delta branch. So why do none of these cracks allow liquid infiltration? I agree with the authors that the gamma grains may be held together by attractive forces from pressure drop prior to liquid infiltration, but as soon as the films are formed I fail to see what forces should hold them together? Why would

the situation be substantially different from the case presented in Fig 2, where detachment is evident? I do not think it suffices to point to the lack of interdendritic flow or convection, or film thickness. I suspect that the lack of detachment can relate to sample confinement (150 micron thick cell) preventing motion of rather thick pieces from the primary trunks. But if this is the case, one cannot exclude that the same confinement/blocking could be partly responsible for crack opening as well? This could explain why visible cracks are not apparent at secondary arms. In summary, I would encourage the authors to substantiate the claim that mechanism number one is a route for fragmentation, or more precisely fragmentation, and detachment in non-constrained 3D geometries. Also, the authors should reflect on the holding times of several minutes required to initiate the process, which seems rather impractical as a protocol to promote grain refinement in real castings.

2) Concerning the second case, i.e. fragmentation caused by a reversed peritectic reaction, the situation is more clear. I find the arguments put forward by the authors in explanation to this case to be rather complete and agree with their conclusions. I do, however, miss some references to previous studies. As far as I can judge, the separation of delta and gamma at gamma grain boundaries by a liquid layer is more or less fully in line with the model proposed by Boussinot et al (Acta Mat 58, 2010, 1750-60), and in more general terms, outside the peritectic system, the process have certain similarities to liquid film migration phenomena (see. e.g Rettenmayr, Intl. Mat. Rev. 54, 2009, 1-17). Finally, just for the sake of completeness, multiple fragmentation along near-isothermal fronts has also been demonstrated in-situ in non-peritectic systems under reversed reactions/remelting, e.g. in dendrites as a consequence of temperature gradient zone melting (Li, et al, Met Mat Trans A37, 2006, 1039-44; Nguyen-Thi, et al, J. Cryst. Growth 310, 2008, 2906-14) and in multicomponent eutectics during constitutionally driven remelting at cell boundaries (Zimmermann, et al, Intl Jou Mat Res 101, 2010, 1484-88).

Some minor issues:

3) Results, page 5, second line, statement: ", which have relatively high spatial resolution," This should be deleted or revised into a (more) meaningful quantitative assessment.

4) Results, page 6, top paragraph: Please clarify whether the sample was held isothermally 20 K below the peritectic temperature after 152 s, or if the temperature gradient was maintained, only stopping the cooling. The difference is not evident from the images alone.

5) Results, page 6, second paragraph: When explaining the dark spots or extinction effects, it is stated that the likelihood for a grain to satisfy the Bragg condition is low due to the high coherency of the incident beam. The use of the term coherency here is unclear and could be misleading as conventional Bragg diffraction with scattering limited to vectors that coincide with the reciprocal lattice is incoherent. I presume the main concerns here is a well- defined Ewald sphere/diffraction condition, resulting from a high degree of collimation (i.e. low divergence) and monochromaticity?

6) Results, page 8, top line: " the diffraction image" should be replaced with " the difference image".

7) Discussion, page 10, bottom paragraph: "Liquid films do not form at the grain boundary of a gamma phase with a carbon content less than C_{δ} because equilibrium between the gamma and liquid phases..." C_{δ} should be replaced with C_{γ} .

Reviewer #2 (Remarks to the Author):

The present manuscript entitled "Dendrite Fragmentation induced by massive-like δ - γ transformation in Fe-C alloys" by Yasuda et al. includes

- new in-situ real time observations of solidification with peritectic transformation in two steel grades,
- new interpretation including remelting at austenite grain boundaries following a massive-like peritectic transformation,
- direct links established between fragmentation and remelting at austenite grain boundaries,
- process conditions for the formation of fragments in peritectic steels.

Strengths of these findings lie in the importance of the newly revealed phenomena for processing of steels with controlled microstructures. The reported experimental data are also to be seen as benchmark for comparison with future modeling efforts that are required to quantify the findings. It is indeed not clear if explanations given by the authors will be confirmed by advanced simulations that couple thermodynamic equilibrium at interfaces in the presence of anisotropic interfacial energy together with kinetics of phase transformations. But the experimental observations presented by the authors are definitively worth sharing with the community.

Charles-Andre GANDIN

Replies to reviewers' comments

The authors greatly appreciate the kind and constructive comments. The authors have carefully considered the comments given by the reviewers and revised the manuscript. Replies to the comments are provided below. The manuscript was revised in accordance with the comments and the replies.

Comments from the reviewers have been italicized and underlined, and numbers in double angle brackets have been added by the authors to correlate the responses to the corresponding reviewer comments.

Reviewer #1 (Remarks to the Author):

The submitted manuscript reports on new dendrite fragmentation mechanisms observed under massive like peritectic transformation and in a reversed peritectic reaction during solidification of Fe-C steels. Such mechanisms would be of major relevance for industries if they could be adapted in to processes promoting grain refinement and improved microstructures in the casting of steels. Although the governing mechanisms may not have been regarded previously as potential candidates to initiate dendrite fragmentation, they are not necessarily novel <<1>>, and I would like to see the present study better linked with previous works <<2>>. I also have some concerns with their potential for industrial applications, as both seem to require some form of interrupted cooling process and holding times, which could turn out very difficult to realize in practice.

As for mechanism no. 1, the massive-like peritectic transformation, I have some concerns with the impact of the sample confinement <<3>>, and would like to see this issue further elaborated in order to judge whether this mechanism truly have detachment potential. For the second mechanism, the reversed peritectic reaction, the case is more evident. Since the two fragmentation mechanisms constitute the core of the submission, I have raised my main criticism in greater detail in the two in bullets 1) and 2) below. Some minor (technical) issues are raised in subsequent bullet points.

1) Concerning the first case, illustrated in Fig.1 and in the supplementary video sequence "melting_GB.mp4": The fragmentation, or crystal fracturing, is apparently initiated by stresses accompanying the massive-like delta-gamma transformation <<4>>. This relates to existing models for high-temperature solidification cracking in Fe-C based alloys, where even non-massive diffusion controlled peritectic transformations have been suggested as a source for internal crack formation, dating back at least 3-4 decades (Brimacombe and Sorimachi, Met. Trans 8B, 1977, 489-505; Clyne, Wolf and Kurz, Met Trans 13B, 1982, 259-266). The authors for the current submission have published results previously, combining in situ radiography and XRD, demonstrating very prominent strain effects associated with massive like transformations (ref. 14 in the manuscript), and I am not convinced that the present case study no. 1 represents something distinctively new. As far as I can tell, a difference in the current situation, compared to the one covered in ref. 14 is the present interruption of the sample cooling, which allows the cracked gamma crystals to be better exposed to liquid. The holding time also allows the system to approach equilibrium and gives time for the carbon transport required to initiate preferential infiltration of liquid into cracks associated

with higher grain-boundary misorientations <<5>>, as clarified through the opening paragraphs of the discussion. The latter bear clear similarities to well-established hot tearing models. In sum, neither the crystal cracking nor the liquid infiltration mechanisms are genuinely new.

The pending issue is therefore if the cracking caused by massive-like delta-gamma transformation can lead to crystal fragmentation, and to me this remains unclear. It is evident both from images and supplementary material that the crystals break up into sub-regions with internal misorientation, evidenced by the extinction effects observed in radiography when separate sub-volumes satisfy the Bragg-condition. Secondly, the rather long incubation time required for liquid infiltration, on a minute scale after the delta-gamma transition, is well explained by the C-diffusion characteristics outlined in the discussion (bottom page 10 and top page 11). Yet, I would expect, if this model was correct, that liquid infiltration should occur first at cracks in secondary arms, where gamma C diffusion completes faster <<6>>. I do not see any evidence in the video that clearly resolvable liquid films are formed in secondary branches, but rather that continuous thicker films are formed more or less exclusively in the primary trunks. At the same time, it is evident that fracturing has happened also inside secondary arms, both from visual extinction, as well as Ostwald ripening events occurring internally inside regions that originated as a single secondary delta branch. So why do none of these cracks allow liquid infiltration? I agree with the authors that the gamma grains may be held together by attractive forces from pressure drop prior to liquid infiltration, but as soon as the films are formed I fail to see what forces should hold them together? <<6>> Why would the situation be substantially different from the case presented in Fig 2, where detachment is evident? I do not think it suffices to point to the lack of interdendritic flow or convection, or film thickness. I suspect that the lack of detachment can relate to sample confinement (150 micron thick cell) preventing motion of rather thick pieces from the primary trunks. But if this is the case, one cannot exclude that the same confinement/blocking could be partly responsible for crack opening as well? This could explain why visible cracks are not apparent at secondary arms. In summary, I would encourage the authors to substantiate the claim that mechanism number one is a route for fragmentation, or more precisely fragmentation, and detachment in non-constrained 3D geometries <<7>>. Also, the authors should reflect on the holding times of several minutes required to initiate the process, which seems rather impractical as a protocol to promote grain refinement in real castings.

<<1>> Novelty of this study

A new finding in this work is that fragmentation (solid grain isolation by liquid film) was induced in a solid grain, which was solidified previously as a delta phase, by massive-like and reverse peritectic transformations. Because the massive-like transformation was confirmed only recently by *in situ* observation (i.e., [Ref 14]), the influences of the massive-like transformation on the microstructure evolution and strain that was induced in the solidifying shell (macroscopic scale) have not been understood sufficiently thus far. Evidence of dendrite fragmentation at the gamma grain boundary and the delta-gamma boundary, which was induced by the massive-like transformation, will provide new and valuable insight towards understanding the fundamentals of solidification in the peritectic systems and the formation mechanism of casting defects, including cracks.

As mentioned at the beginning of the “Discussion”, a relatively large interfacial energy between the solid and liquid phases often impedes bonding between the solid grains even at a high solids fraction. Therefore, in the solidification without any solid-state transformation, solid grains with different crystallographic orientations are isolated by the liquid phase and the grain boundaries are not formed until the final stage of solidification. Because the remaining liquid at the grain boundary has been studied separately, the authors did not intend to emphasize this information in this paper. Instead, the focus of this paper is on fragmentation that is induced by the massive-like transformation.

<<2>> Previous studies

In a previous publication ([Ref. 14]), the authors showed that massive-like transformation (solid-state transformation from the gamma to the delta phases) could be selected. The paper ([Ref. 14]) presents two different cases: one is transformation in the mushy region during delta solidification, and the other is transformation after delta solidification has been completed (100 K below the peritectic temperature, in the single austenite region). The second case was defined as the massive-like transformation. However, we did not confirm whether massive-like transformation occurred in the first case. Thus, the transformation from the delta to the gamma phases in the mushy region was categorized into the peritectic mode in the paper. Recently, we proved that the massive-like transformation was selected even at low growth velocities (10 $\mu\text{m/s}$) during unidirectional solidification [Ref 16, English version will be submitted to ISIJ International]. Those studies focused on the selection of the massive-like transformation.

<<3>> Impact of sample confinement

(a) Selection of massive-like transformation

The authors agree with the reviewer that the thermal and solutal fields were influenced by the sample confinement (thickness: 100 μm in this study), and consequently the microstructure evolution could be modified. In addition, solid grain motion (including fragmented grains) is restricted as shown in Fig. C1. The solid grains can be piled up in thin space. Thus, it is plausible to assume that confinement can hinder, rather than promote, the detachment of fragmented arms. However, the selection of the massive-like transformation and the induced fragmentation were not influenced by the confinement.

(b) Contact of solid grains with the sapphire plates (sample holder)

The interfacial energy of the solid–liquid phases for pure iron, $\sigma_{\text{Solid-Liquid}}$, was estimated to be 0.2 J/m^2 [R01]. The value of $\sigma_{\text{Al}_2\text{O}_3\text{-Solid}} - (\sigma_{\text{Solid-Liquid}} + \sigma_{\text{Al}_2\text{O}_3\text{-Liquid}})$ was calculated to be +0.2 J/m^2 [R02]. In terms of interfacial energies, the positive value indicates that the solid phase does not stick (bond) to the sapphire plates during solidification (see Fig. C1). Therefore, the friction between the solid grain and the sapphire plate is expected to be relatively small. The detachment of dendrite arms as shown in Fig. 3 indicates that the dendrites did not stick to the sapphire plates. An *in situ* observation of the semisolid deformation in steel [Ref 13]) showed that solid grains moved / rotated and solid grain rearrangement played a dominant role in this behavior. Therefore, we concluded that the friction force does not influence the strain during and after the massive-like transformation. Any tensile stress that expands the gap between

the solid grains was not induced by sample confinement.

Fig. C1. Pile-up of solid grains in thin sample holder.

(c) Locally induced strain by massive-like transformation and role of liquid

As the reviewer pointed out, one of the authors (H.Y.) reported the large strain that was induced by the massive-like transformation in [Ref 14]. According to the distorted X-ray Laue spots (white X-rays with a beam size of $0.3 \text{ mm} \times 0.3 \text{ mm}$), the divergence angle was as large as 10° in a single gamma grain. Figure C2 shows a schematic illustration of the crystal lattice that was distorted by the massive-like transformation. The lattice distortion resulted in the divergence of Laue diffraction spots. As shown in the Fig. C2, the crystal lattice was distorted locally and the spots did not show any indication of uniaxial tensile strain.

Fig C2. Schematic illustration of crystal lattice. Left: before the massive-like transformation (delta phase), right: distorted lattice (gamma phase) observed by X-ray diffraction.

According to observation (Fig. 1 of this paper), gamma-grain coarsening occurred in dendrite arms (primary and the secondary arms) immediately after the massive-like transformation because of the high mobility of atoms at the high temperatures. The

coarsening, in which atomic configuration was rearranged to reduce the free energy of the system, eliminated the elastic strain and reduced the plastic strain. In addition, the dendrite arms did not stick to the sapphire plates as shown in Fig. C3 (explained above). The formation of multiple gamma grains with different crystallographic orientations did not simply generate uniform plastic strain.

Fig. C3. Schematic illustration of fragmented arms.

In conclusion, based on (a)–(c), the liquid film that was produced at the gamma grain boundary was attributed to the relatively large grain boundary energy compared with the solid–liquid interfacial energy. The tensile strain which induced fragmentation and detachment was not generated within the dendrite arms.

<<4>> Strain for considering cracking/hot tearing of solidifying shell

Cracking/hot-tearing in large castings (i.e., the slab in CC) has been discussed in terms of induced strains, as mentioned by the reviewer and in the literature [R03][R04]. Here, the strain should be defined as a macroscopic strain. The strain is induced by various mechanisms (friction between the solidifying shell and mold, mold oscillation, static pressure, machine misalignment, bending and straightening). Volume shrinkage from the delta–gamma transformation has to be included as a source of the induced strain. The local strain that is induced by the massive-like transformation [Ref 14] should be included, although it can be done only in a non-straightforward way as discussed in more detail below.

As mentioned above, the local strain is induced within the gamma grains. Figure C4 shows a schematic illustration of the solidifying shell. Because the contact between the solidifying shell and the mold through the slag (mold flux) is not uniform, the temperature field is not uniform. During slab cooling (with a typical width of 1–2 m) in the mold, the massive-like transformation will begin at various surface region positions. As a result, the solidifying shell shrinks non-uniformly, depending on the frequency of gamma-phase nucleation events (spatial and temporal distribution of gamma-phase nucleation). Thus, the generation of macroscopic strain should not be considered by a single massive-like transformation event.

A critical strain has been proposed to predict the formation of internal cracking in the continuous casting [R05][R06]. According to those studies, the critical strain is considered a function of the strain that is induced in the brittle temperature range, the strain rate and the carbon content. The critical strain, which was measured experimentally, ranged from 1% to 4%, depending on the strain rate and the carbon content. Recently, volume shrinkage in the massive-like transformation was measured by time-resolved *in situ* tomography [R07]. The strain that was induced by the isotropic shrinkage because of the massive-like transformation was estimated to be only –0.17 %, because the volume change from the massive-like transformation was –0.5 %. The

critical strain for the internal crack formation is significantly larger than the local strain that was induced by the massive-like transformation. The difference between the critical strain and the local strain indicates clearly that the massive-like transformation should be included as one of the sources of macroscopic strain (non-uniform strain).

Although the authors are interested in the macroscopic strain that is induced in the solidifying shell (Fig. C4), we intend to focus on fragmentation in this paper. A brief discussion was added in the revised manuscript.

Fig. C4 Schematic illustration of solidifying shell in continuous casting.

<<5>> Holding time for fragmentation

(a) Formation of thin liquid film immediately after massive-like transformation

As presented in this paper, it took minutes to observe the thick liquid film at the gamma grain boundary. The authors agree with the reviewer that a holding time of minutes appears long for practical application. However, the fragmentation observed in this paper is still of interest for developing practical applications.

The liquid film was formed immediately after the massive-like transformation. The gamma grain boundary was identified immediately after the massive-like transformation when the massive-like transformation occurred while keeping the temperature below the peritectic temperature, as shown in Fig. 1 and in the attached movie. If the liquid phase is not produced at the grain boundary, the grain boundary cannot be detected because the difference in the X-ray absorption coefficient between the inside and boundary of the grains is extremely small. In a previous paper [Ref 14], the grain boundary of gamma grains (liquid film) was never observed after the massive-like transformation when the specimen was cooled at a constant cooling rate from 5 K/min to 30 K/min. Thus, to keep the temperature below the peritectic temperature or to reduce the cooling rate significantly below the peritectic temperature is required to produce liquid film in the gamma grains.

According to Fig. 1, the liquid film thickness could be 1–2 μm or even less (4 pixels in the transmission images) immediately after the massive-like transformation. Because the thickness is nearly equal to the spatial resolution, it was difficult to measure the thickness precisely. In the original manuscript, an explanation of the thin liquid film was insufficient. We revised the manuscript to explain the formation of a thin liquid film.

(b) Method to use or to avoid intentional fragmentation

The use or avoidance of fragmentation that is induced by the massive-like transformation will be the focus of research and development in the next step. This paper explains fragmentation that is induced by the massive-like transformation to share knowledge on fragmentation with the community and to provide motivation for future work.

Although details of possible industrial applications need to be explored, we can discuss some links of fragmentation to practical processes.

Case 1: Stirring and secondary cooling

In continuous steel casting, casting is not cooled at a constant rate. For instance, electromagnetic stirring increases the heat flux from the melt to the solidifying shell and consequently, the cooling rate is reduced significantly or even becomes negative. This is one of the preferred conditions for fragmentation.

Grain refinement is often achieved by stirring in steel castings. If dendrites at the advancing front are fragmented and detached by simple remelting from stirring, fragmentation is expected to occur over a wide range of carbon concentrations. However, grain refinement was achieved only in a certain composition range (high carbon content, i.e., 0.45 mass% C). The position of the delta–gamma interface depends on the carbon concentration. It is worth studying the influence of the massive-like transformation on the grain refinement. The massive-like transformation can influence the microstructure formation in practical processes even if we did not notice the massive-like transformation itself.

A reduction in cooling rate occurs at other positions during continuous casting. For instance, the cooling rate decreases when the casting moves out of the mold and increases again in the secondary cooling zone. Usually, the cooling rate fluctuates in the secondary cooling zone.

Case 2: Deformation of semisolid (gamma + melt)

The formation of a liquid film at the gamma grain boundary resulted in the semisolid state. The semisolid exhibits a unique deformation behavior. For instance, the rearrangement of solid grains plays a dominant role in deformation, which leads to the localization of shear deformation, segregation bands, and surface and internal cracking. In continuous casting, various mechanisms (such as mold oscillation, rolling, bending) cause stress/strain in the solidifying shell [R03][R04]. The rearrangement of fragmented gamma grains can initiate surface and internal cracking under a certain external force even if the liquid film thickness is thin (in the order of μm). It is also worth studying the influence of the massive-like transformation on the deformation behavior of the solidifying shell.

We revised the manuscript to make these links clear.

<<6>> Fragmentation of secondary arms

In the movie, at 355 s, only a few grain gamma-phase boundaries were found in the secondary arms. The actual number of grain boundaries in the secondary arms could be greater. Most grain boundaries with a thin liquid film moved into the liquid phase during coarsening (most gamma grains that were produced by the massive-like transformation were eliminated) in the secondary arms during coarsening. As a result, the liquid film was not observed frequently in the secondary arms. The liquid film that was caused by grain boundary melting occurred in the secondary and primary arms.

Once the liquid film with a certain thickness had formed at the grain boundary of the gamma phase (Fig. 1), no driving force for further melting existed because the total interfacial energy did not depend on the liquid film thickness.

The reverse peritectic reaction produced the liquid and delta phase from the gamma phase (Fig. 2), which lead to the formation of a thick liquid film at the delta–gamma boundary. According to the modeling of an interaction between particles and the solid–

liquid interface [R08], the viscous drag force is inversely proportional to the gap between the particles and the solid–liquid interface. The viscous drag force that operates at the delta–gamma boundary may become small, and consequently, detachment occurs easily in the reverse peritectic reaction.

<<7>> Three-dimensional observation

Although the authors believe that the results and discussion are sufficient for the context and contents of this article, the authors agree with the reviewer in that three-dimensional observation would reveal more than what we could have revealed. As a result, the authors examined the possibilities of three-dimensional (3D) observation after receiving the reviewer’s comments by using beamtime at the SPring-8 (synchrotron radiation facility) in 2019 to ensure that the replies to the reviewer comments are as credible as possible. Furthermore, the authors identified possible next steps in future studies by identifying necessary technological conditions that are required for 3D observation, compared with those available at one of the most advanced beamlines with beamline scientists allowing flexible and alternative choices for advanced measurements, that is, not confined to de facto standards at the beamline.

We have developed time-resolved *in situ* tomography (4D-CT) for steel solidification. Compared with Al–Cu alloys, the contrast resolution between the solid and liquid phases in carbon steel is much lower [Ref 12] which made it difficult to perform the 3D observation of dendrites especially in steel using currently available technological resources. As shown in Fig. C5, 4D-CT and an originally developed image-processing procedure have enabled the observation of dendrites of carbon steel (Fe-0.45C-0.6Mn-2Si) *in situ*. In this state, the voxel size is $6.5 \mu\text{m} \times 6.5 \mu\text{m} \times 6.5 \mu\text{m}$ and a 360° rotation takes 4 s.

Fig. C5. 3D observation of dendrites in carbon steel.

We examined the required conditions to observe the fragmentation phenomena related to massive-like transformation by X-ray tomography. The requirements are:

- * A spatial resolution as small as $1 \mu\text{m}$ to observe the liquid film of which the thickness is of the order of μm .
- * A temporal resolution of 4D-CT as short as 1 s or less to observe grain boundary motion (liquid film). Even in the late stage of coarsening, the time resolution should be as short as 10 s.
- * Monochromatized X-rays are needed to obtain sufficient contrast resolution between the solid and liquid phases (a pink X-ray beam improves temporal resolution significantly, but degrades the contrast resolution).

In the current 4D-CT setup using the monochromatized X-ray at BL20XU, which is

the most advanced and suitable beamline in SPring-8 for our observations, at least 20–60 s was required for every rotation to achieve a spatial resolution of 0.5–3 μm . In addition, the sample size (diameter) should be smaller than 0.5 mm. Spatial resolution becomes as large as 7.5–12 μm when a 360° rotation takes 1 s (temporal resolution is 1s).

A pink X-ray beam (X-rays with high energies are cut by a mirror; the total X-ray intensity is much higher than that of the monochromatized X-ray, available at the BL28B2 of SPring-8) is an alternative to improve the temporal resolution. However, we have confirmed experimentally that the contrast resolution between the solid and the liquid phases in the Fe-C alloys became too low to observe the solidification structure, especially for the liquid film at the delta–gamma grain boundaries.

We conclude that further technical innovations of X-ray sources and/or beam monitors are required to observe fragmentation in carbon steel. We believe that even without 3D observations, results that are obtained by transmission imaging (2D observation) are novel and valuable for understanding the solidification phenomena in carbon steel, because fragmentation that is induced by the massive-like transformation has not been considered to date. The 3D observation will be conducted in future.

2) Concerning the second case, i.e. fragmentation caused by a reversed peritectic reaction, the situation is more clear. I find the arguments put forward by the authors in explanation to this case to be rather complete and agree with their conclusions. I do, however, miss some references to previous studies. As far as I can judge, the separation of delta and gamma at gamma grain boundaries by a liquid layer is more or less fully in line with the model proposed by Boussinot et al (Acta Mat 58, 2010, 1750-60), and in more general terms, outside the peritectic system, the process have certain similarities to liquid film migration phenomena (see. e.g Rettenmayr, Intl. Mat. Rev. 54, 2009, 1-17). Finally, just for the sake of completeness, multiple fragmentation along near-isothermal fronts has also been demonstrated in-situ in non-peritectic systems under reversed reactions/remelting, e.g. in dendrites as a consequence of temperature gradient zone melting (Li, et al, Met Mat Trans A37, 2006, 1039-44; Nguyen-Thi, et al, J. Cryst. Growth 310, 2008, 2906-14) and in multicomponent eutectics during constitutionally driven remelting at cell boundaries (Zimmermann, et al, Intl Jou Mat Res 101, 2010, 1484-88).

We have included arguments on the liquid film formation and fragmentation from melting and cite related papers, including the papers suggested by the reviewer [R09][R10].

Some minor issues:

3) Results, page 5, second line, statement: “, which have relatively high spatial resolution,” This should be deleted or revised into a (more) meaningful quantitative assessment.

We have deleted the words.

4) Results, page 6, top paragraph: Please clarify whether the sample was held

isothermally 20 K below the peritectic temperature after 152 s, or if the temperature gradient was maintained, only stopping the cooling. The difference is not evident from the images alone.

Cooling was suspended and the temperature gradient was maintained. We included an explanation on the temperature gradient.

5) Results, page 6, second paragraph: When explaining the dark spots or extinction effects, it is stated that the likelihood for a grain to satisfy the Bragg condition is low due to the high coherency of the incident beam. The use of the term coherency here is unclear and could be misleading as conventional Bragg diffraction with scattering limited to vectors that coincide with the reciprocal lattice is incoherent. I presume the main concern here is a well-defined Ewald sphere/diffraction condition, resulting from a high degree of collimation (i.e. low divergence) and monochromaticity?

Thank you for the comment. As suggested by the reviewer, the term “coherency” can be misleading. We have modified the explanation. In the beamline BL20XU of the Spring-8, the X-ray divergence is small (the distance between the X-ray source and the specimen was ~240 m) and the X-ray energy width was also small (monochromator using double Si crystals).

6) Results, page 8, top line: “ the diffraction image” should be replaced with “ the difference image”.

We have corrected the words.

7) Discussion, page 10, bottom paragraph: “Liquid films do not form at the grain boundary of a gamma phase with a carbon content less than C_{δ} because equilibrium between the gamma and liquid phases...” C_{δ} should be replaced with C_{γ} .

We have corrected the words.

Reviewer #2 (Remarks to the Author):

The present manuscript entitled “Dendrite Fragmentation induced by massive-like δ - γ transformation in Fe-C alloys” by Yasuda et al. includes

- new in-situ real time observations of solidification with peritectic transformation in two steel grades,*
- new interpretation including remelting at austenite grain boundaries following a massive-like peritectic transformation,*
- direct links established between fragmentation and remelting at austenite grain boundaries,*
- process conditions for the formation of fragments in peritectic steels.*

Strengths of these findings lie in the importance of the newly revealed phenomena for processing of steels with controlled microstructures. The reported experimental data are

also to be seen as benchmark for comparison with future modeling efforts that are required to quantify the findings. It is indeed not clear if explanations given by the authors will be confirmed by advanced simulations that couple thermodynamic equilibrium at interfaces in the presence of anisotropic interfacial energy together with kinetics of phase transformations. But the experimental observations presented by the authors are definitively worth sharing with the community.

The authors appreciate the reviewer's comments. As indicated by the reviewer, the formation of a liquid film at the grain boundary should be influenced by the grain boundary energy anisotropy. We started to develop 4D-CT and 3DXRD techniques in which the solid grain configuration (4D-CT) and crystallographic orientation of each grain (3DXRD) were measured simultaneously. We will aim to present the results in our future publications.

References

- [R01] Turnbull, D. Correlation of Liquid - Solid Interfacial Energies Calculated from Supercooling of Small Droplets. *J. Chemical Physics*, **18**, 769 (1950).
- [R02] Shibata, H., Yin, H., Yoshinaga, S., Emi, T., Suzuki, M., In-situ Observation of Engulfment and Pushing of Nonmetallic Inclusions in Steel Melt by Advancing Melt / Solid Interface. *ISIJ International*, **38**, 149 (1998).
- [R03] Brimacombe, J. K. & Sorimachi, K., Crack Formation in the Continuous Casting of Steel. *Metall. Trans. B*, **8B**, 489 (1977).
- [R04] Clyne, T. W., Wolf, M. & Kurz, W., The effect of Melt Composition on Solidification Cracking of Steel, with Particular Reference to Continuous Casting. *Metall. Trans. B*, **13B**, 259 (1982).
- [R05] Yamanaka, A., Nakajima, K. & Okamura, K., Critical strain for internal crack formation in continuous casting, *Iron and Steelmaking*, **22**, 508 (1995).
- [R06] Won, Y. M., Yeo, T. J., Seol, D. J. & Oh, K. H., A New Criterion for Internal Crack Formation in Continuously Cast Steels, *Metall. Mater. Trans. B*, **31B**, 779 (2000).
- [R07] Yasuda, H., Hashimoto, T., Sei, N., Morishita, K. & Yoshiya, M., Investigation using 4D-CT of massive-like transformation from the δ to γ phase during and after δ -solidification in carbon steels, *IOP conf. ser.: Mater. Sci. Eng.* (in press).
- [R08] Uhlmann, D. R., Chalmers, B. & Jackson, K. A. Interaction Between Particles and a Solid-Liquid Interface. *J. App. Phys.* **35**, 2986 (1964).
- [R09] Boussinot, G., Brener, E. A. & Temkin, D. E. Kinetics of isothermal phase transformations above and below the peritectic temperature: Phase-field simulations. *Acta Mater.* **58**, 1750-60 (2010).
- [R10] Rettenmay, M. Melting and remelting phenomena, *Int. Mater. Rev.* **54**, 1 (2013).

REVIEWERS' COMMENTS:

Reviewer #1 (Remarks to the Author):

The authors have provided detailed and thorough responses to all comments and issues raised by me in the first review. I find their revisions and replies to be both satisfactory and enlightening, and believe clarifications to the pending issues has improved the manuscript quality.